# Evaluating case management as a complex intervention: Lessons for the future

**Anne-Sophie Lambert**[1]*, **Catherine Legrand**[2], **Sophie Cès**[1], **Thérèse Van Durme**[1], **Jean Macq**[1]

**1** Institute of Health and Society (IRSS), Université Catholique de Louvain, Clos Chapelle aux Champs, Brussels, Belgium, **2** Institute of Statistics, Biostatistics and Actuarial Sciences (ISBA-IMMAQ), Université Catholique de Louvain, Louvain-la-Neuve, Belgium

* anne-sophie.lambert@uclouvain.be

## Abstract

The methodological challenges to effectiveness evaluation of complex interventions has been widely discussed. Bottom-up case management for frail older person was implemented in Belgium, and indeed, it was evaluated as a complex intervention. This paper presents the methodological approach we developed to respond to four main methodological challenges regarding the evaluation of case management: (1) the standardization of the interventions, (2) stratification of the frail older population that was used to test various modalities of case management with different risks groups, (3) the building of a control group, and (4) the use of multiple outcomes in evaluating case management. To address these challenges, we developed a mixed-methods approach that (1) used multiple embedded case studies to classify case management types according to their characteristics and implementation conditions; and (2) compared subgroups of beneficiaries with specific needs (defined by Principal Component Analysis prior to cluster analysis) and a control group receiving 'usual care', to evaluate the effectiveness of case management. The beneficiaries' subgroups were matched using propensity scores and compared using generalized pairwise comparison and the hurdle model with the control group. Our results suggest that the impact of case management on patient health and the services used varies according to specific needs and categories of case management. However, these equivocal results question our methodological approach. We suggest to reconsider the evaluation approach by moving away from a viewing case management as an intervention. Rather, it should be considered as a process of interconnected actions taking place within a complex system.

## Background

Like many countries, Belgium currently has an increasing number of frail older people [1], many of whom are highly vulnerable to 'adverse health outcomes, including disability, dependency, falls, long-term care need and mortality [2].' Given the difficulties inherent not only to rapidly finding nursing home beds [3], but also to these people's living preferences, many of these elderly people are likely to live at home longer with the support of family carers and the

**Data Availability Statement:** Under the recommendations of the Belgian Privacy Commission, the datasets generated and/or analysed during the current study are not publicly available. However, any person can apply for

access to the data through the National Institute of Health and Disability Insurance (NIHDI) (the data owner) and the Belgian Privacy Commission. The contact person is Daniel Crabbe, Daniel. crabbe@riziv.fgov.be.

**Funding:** This study is part of Protocol 3, a scientific evaluation of alternative care interventions that aim to prevent the risk of institutionalization and maintain a satisfactory quality of life. The intervention and scientific evaluation were financed by the National Institute of Health and Disability Insurance (NIHDI).

**Competing interests:** The authors have declared that no competing interests exist.

**Abbreviations:** NIHDI, National Insurance for Health and Disability Institute; CM, Case Management; GP, General Practitioners; ADL, Activities of Daily Life; IADL, Instrumental Activities of Daily Life; CPS, Cognitive Performance Scale; DRS, Depression Rating Scale; MCA, Multiple Correspondence Analysis; PCA, Principal Component Analysis; IADL (cogn.), IADL and low level of cognitive impairments; Func., cogn., Functional & cognitive; Func., cogn., behav., Functional, cognitive & behavioural problems; GPC, Generalized Pairwise Comparison; Δ, net benefit; ZAP, Zero-Augmented Poisson model; ZANB, Zero-Augmented Negative Binomial model; 95% CI, 95% confidence interval; OR, Odds Ratio; RR, Relative Risk.

formal health and social care services. Within the last decade, some reforms have been implemented to support this population in the setting of a health system that is still hospital-centred and is oriented towards acute care [4].

Part of this reform process has involved the implementation of innovative homebound interventions (such as community-care case management, psychological support, occupational therapy, and night care), whose initial aim was to support frail older people at home and to prevent definitive institutionalization. As components of a programme labelled Protocol 3, these interventions were designed bottom-up and 75 projects were implemented from 2010 to 2016 in the form of pilot projects. The programme was financed by the National Institute of Health and Disability Insurance (NIHDI), which, together with policymakers, asked a consortium of universities to evaluate these pilot projects. The main research questions were [5]: What are the bottom-up interventions all about, who should benefit from them, what is the range of intended effects and how are they likely to vary among sub-groups of recipients?

This paper focuses on the 50 projects that implemented Case Management (CM), which is known to contribute to care integration [6, 7]. CM is defined as 'a collaborative process of assessment, planning, facilitation, care coordination, evaluation, and advocacy for options and services to meet an individual's and family's comprehensive health needs through communication and available resources to promote quality, cost-effective outcomes' [8]. To be labelled as CM, the submission files for Protocol 3 projects had to present at least three of the six features included in this definition.

Evaluating the effectiveness of CM requires a specific approach that was developed for the evaluation of complex interventions [9, 10]. This is because CM can indeed be viewed as a complex intervention: it has many interacting components, requires a wide variety of behaviours or actions on the part of those who deliver or receive it, targets various organizational levels, has very diverse outcomes, and requires considerable flexibility in its tailoring [11]. If CM is evaluated according to a classical methodological approach, there are therefore four main challenges: (1) the standardization of the interventions in a recognizable form [12], (2) the stratification of the study population in a way that adapts an appropriate care model to match the patients' specific needs within groups [13, 14], (3) building a comparison group [15], and (4) evaluating the effectiveness of CM with a broad view of the different intervention impacts, including those related to clinical outcomes, patient experiences and utilization of services [16].

The aims of this paper are thus (1) to present and discuss the methodological choices we made to address these four main challenges; (2) to present the results of this evaluation, and (3) to discuss the lessons learned from it by reflecting on a possible shift away from evaluating CM as complex interventions and towards its evaluation as a process of interconnected actions within a complex system.

## Methods

### Study design

To address the methodological challenges listed above, we used a mixed-methods design with concurrent triangulation [17], a process that was intended to describe the interventions, identify facilitators and barriers to their implementation, and assess their impact on clinical outcomes and on service utilization for different sub-groups of frail older people.

To address context barriers leading to the identification of a CM classification, the qualitative sub-study (a multiple embedded case study) focused on the precise description of interventions [7]. The CM classification's objective is to standardize the projects that have been included in the same category [18].

The quantitative sub-study (an observational longitudinal design) started by identifying sub-groups of beneficiaries of similar types and with similar levels of need. The impact on clinical outcomes and services utilization of each CM category that had been highlighted in the qualitative sub-study was then evaluated for each sub-group of the population [19].

## Inclusion criteria for target population

The patients included in this study–the intervention and control groups–met the following inclusion criteria [5]: living at home at baseline and being at least 60 years old; and either scoring at least six on the Edmonton Frail Scale [20], or having either a dependence status of A, B, or C on the Katz home scale, or of B, C, or Cd on the Katz residential scale [21]; or having been diagnosed with dementia by a geriatrician, neurologist or psychiatrist.

## Qualitative methodology

**Development of a standardized forms of CM.** The classification of CM projects started by the building of a normative grid: a list which captures what the stakeholders of the projects believed to be the key components of the projects [22]. This list was built using data from six in-depth cases studies, selected among the 50 projects that implemented CM, to extract as much diversity as possible regarding the projects 'characteristics [23,24, 25]. These six case studies were thus selected on the basis of (1) their diversity (i.e. the profile of the case manager (i.e. nurse, occupational therapist, psychologist or social worker)); (2) their geographical location (urban or rural); and (3) their collaboration with an existing organization that, among other criteria, coordinated care. The methodology of this part of the study has been described elsewhere [7]. Yearly data collection included submission files, questionnaires, routinely produced project documents, and focus-group and individual semi-structured interviews. The latter targeted a variety of project stakeholders, including professionals, frail older people and their family carers. This produced a list of 23 components evaluated with a maximum of four levels of achievement [7], organized along the six domains of the Chronic Care Model [26] and validated by a group of experts and stakeholders who were working on the pilot projects [27].

After that, the 50 projects identified as CM from their submission files were evaluated on the basis of the four achievement levels defined for the 23 components. To highlight the main dimensions of CM, the third step conducted a Multiple Correspondence Analysis (MCA) on these data [7]. The level of feedback provided to the frail older people' general practitioner was the most weighted component in our study and was considered to be a proxy for structured inter-professional work.

Another important determinant of the effectiveness of CM, according to the literature, is based on the intensity of the intervention [28]. In our study, the intensity was operationalized as the average monthly number of home visits performed by the case manager. A frequency lower than two home visits per month indicated a low intensity and a frequency of two home visits per month or higher indicated a high intensity [29].

In this way, three intervention categories emerged from the analysis. Although the first, 'basic care coordination' projects, were scheduled to provide CM in the submission file, they did not implement the main component of the CM (i.e. feedback with the GP). This category grouped 7 projects in which 324 frail older persons are included (S1 Table). The second, 'low-intensity CM' projects, implemented the main component of the CM at a frequency of less than two home visits per month. It grouped 16 projects in which 859 frail older persons are included (S1 Table). Finally, 'high-intensity CM' projects implemented the main component

of the CM at a frequency of two home visits or more per month. It grouped 27 projects in which 1,833 frail older persons are included (S1 Table).

## Quantitative methodology

**Data collection.** To collect the quantitative data, two databases were merged using national registration numbers. The first database contained prospective data on the main clinical variables contained in the InterRAI-HC, a web-based comprehensive geriatric assessment tool that has been adapted for use in Belgium (BelRAI-HC). It includes the scales for evaluating the following: functional performance of Activities of Daily Living (ADL), measured on the ADLH scale; functional performance of Instrumental Activities of Daily Living (IADL), measured by IADLp; cognitive performance (CPS), measured on the CPS2; depression, measured on the Depression Rating Scale (DRS); and behavioural problems, measured on the Aggressive Behaviour Scale (ABS). The BelRAI-HC database also includes variables to define the extent to which the main family carer was present; this was coded according to the living arrangements as a categorical variable at three levels: without a family carer, with a non-cohabitant family carer, or with a cohabitant family carer. For Protocol 3, two scales were added to the BelRAI-HC: the 12-item version of the Zarit-Burden interview (ZBI-12) (to estimate the perceived psychological burden on the family carer) [30]; and the WHO-QoL-8 (to measure the patient's perceived quality of life [31]). Professionals of projects encoded their observations in the BelRAI-HC at enrolment, at the patients' exit from the project, and/or six months after enrolment[19].

The second database, an administrative database on reimbursed healthcare consumption, was available from one year before inclusion and at six months follow-up. These two merged databases were cleaned such that they only contained complete evaluations at baseline and six months after enrolment. Data were available for 3,016 beneficiaries of CM (intervention group), and 552 older people benefitting from usual care, who had been recruited at home by nursing-care organizations (control group).

**Ethical approval and informed consent.** The use of the data for this study was approved by the Belgian Privacy Commission and by the Ethics Committees at both the Université Catholique de Louvain and the Katholieke Universiteit Leuven under dossier number B4032010 8337. All study participants are asked to sign an informed consent agreement prior to participating in the study. However, if a participant is not capable of signing the document, a family member or his legal representative will sign it on their behalf, as stipulated by the Belgian law [32]. All data are anonymized and analysed following the rules of the Belgian Privacy Commission [33].

**Stratification of the population.** Despite the inclusion criteria described above, the population included in the pilot projects was heterogeneous with regard to its manifestations of frailty and its long-term care needs. Although the literature suggests that disability profiles are the best determinant for identifying long-term care needs [34], the need for case management is defined by more than the care receiver's disability profile. Even when the disability profile is severe, it is possible–when care providers can align the needs and wishes of the care receiver efficiently and flexibly–for care and support for care to be well-aligned with the care needs of the care receiver [35]. However, when care needs and care provision are misaligned–for instance, when social and care support are is ill-coordinated–case management by a professional can provide considerable added value [35].

Unfortunately, it was not possible to stratify the population on the basis of these alignment (or misalignment) data. This is why we had to rely solely on disability profiles. To determine the population subgroups with specific long-term care needs, we therefore identified 'natural'

clustering of individuals with similar disability profiles in the intervention group. For this, prior to cluster analysis, we used the 'FactoMineR' package in R software [36] to perform a Principal Component Analysis (PCA) on the different clinical scores (including IADL, ADL, CPS and ABS) [37]. The number of groups highlighted by the use of a hierarchical algorithm (Ward algorithm) is the compromise between intra-group homogeneity and inter-group heterogeneity.

Five disability profiles emerged from this analysis [15]. The present paper focuses on profiles with cognitive impairments: (1) people with IADL limitations and slight cognitive impairments (IADL, (cogn.)) (N = 1,930) (S1 Table); (2) people with functional and cognitive impairments (Func., cogn.) (N = 840) (S1 Table); and (3) people with functional, cognitive and behavioural problems (Func., cogn., behav.) (N = 246) (S1 Table). These profiles are considered to be the main target population for CM [38, 39], as frail older people with cognitive impairments are likely to have increased levels of disability, to be at high risk of misalignment between care needs of care receiver, and therefore to require constant adaptations for support from a variety of social and healthcare services [28].

Descriptions of the different disability profiles were first presented as the proportion of patients with a clinical score above the cut-off. The cut-off are defined as a total score greater than 3 for ADL [40], CPS [41] and DRS [42], and greater than 24 for IADLp indicating significant limitations, a total score greater than 0 for ABS indicating behavioural problems and a total score greater than 10 on the Zarit scale [30] which is associated with a higher risk for depressive symptoms among family carers.

Second, the proportion of different service users was described for each disability profile. The use of personal nursing care was described as a binary variable, with zero being unjustified use and one being justified use. 'Justified personal nursing care' was coded when a person *with* significant difficulties in performing hygiene tasks received personal nursing care. It was also used in cases when a person *without* significant difficulties in performing hygiene tasks did *not* receive personal nursing care. Difficulties in performing hygiene tasks were assessed on the basis of an item on the ADL hierarchy scale: 'the ability to wash oneself using a sink'. On the other hand, 'unjustified use of personal nursing care' was coded when a person *with* significant difficulties in performing hygiene tasks did *not* receive personal nursing care. This code was also used when a person *without* significant difficulties in performing hygiene tasks received nursing personal care. The use of respite services–a high-value service–was described as the proportion of short-term institutionalizations (less than 90 consecutive days) and the proportion of day-care centre use. The proportions of emergency-department service use and general-practitioner (GP) visits in the evening, at night or during public holidays (GP out-of-hours visits) were described as low-value services, on the principle that much of the use of these services could have been avoided if there had been adequate support at home.

**Creating a control group.** The intervention group was compared to a group that benefited from 'usual care' (control group). The control group was constituted first using a stratification that established similarities with regard to the presence levels of family carers, and then, per disability profile, by performing one-to-one propensity-score matching (with binary dependent variable intervention versus control group) based on disability scales (ADL, IADL, CPS, DRS and, ABS), age, and gender [15]. The matching function used was one-to-one matching of the nearest neighbours with replacement [15] and was computed using the R package 'matching'[43].

With one exception, the matching balance of multiple covariates evaluated for each disability profile on the basis of the average standardized mean difference and the geometric mean variance ratio did not show significant differences. The exception concerned older people who did not have a family caregiver but had functional and cognitive impairments or functional,

cognitive and behavioural problems (see S2 Table). This can be explained by the low number of older people–representing only 5% of the people included in these profiles–who had no family caregiver but lived at home with significant cognitive impairments[15].

**Outcome evaluation.** The outcomes included in this study are related to clinical outcomes and the use of services. Regarding the latter, we hypothesized that CM reduces the use of low added-value services (i.e. the avoidable use of emergency departments and out-of-hours GP visits), while also increasing the use of high added-value services (i.e. personal nursing care or respite services). It should be noted that we did not consider definitive institutionalization as a relevant endpoint, since, depending on a patient's health and social context, it is sometimes a suitable solution [44, 45]. In such situations, the case manager–whose role is to align care needs and care receivers–will prepare such a person for institutionalisation. This was not therefore evaluated in the present analysis. In addition, as statistical analysis was impeded by the low proportions of users, only descriptive statistics were computed for the use of respite services.

Clinical outcomes and the proportion of justified use of personal nursing care were evaluated using a statistical technique known as Generalized Pairwise Comparisons (GPC) [46]. For the variable of interest, this analysis compared every possible pair of patients between the intervention and control groups, and determined whether each pair was 'favourable' (i.e. that the outcome of the patient in the intervention group was 'better' than of the patient in the control group); 'unfavourable' (i.e. the outcome of the patient in the intervention group was 'worse' than that of the patient in the control group); or 'neutral' (i.e. there was no difference between the outcomes of the two patients). Given a pre-specified threshold 'τ,' a pair was considered to be favourable if the difference between the clinical scores of the patient in the intervention group and of the patient in the control group was larger than τ, and as unfavourable if it was lower than -τ. Pairs for which the difference was between [-τ and τ] were considered to be neutral. To interpret the results as described above, the clinical scales were inverted (i.e. the maximum score corresponded to the situations in which a patient had no limitations and the minimum score corresponded to the situations in which a patient had full limitations).

The pre-specified threshold of the ADL scale was fixed at one, indicating a possible change in a patient's need for healthcare services. For the IADL scale, each activity was dichotomized to disabled or non-disabled, and summed to obtain a total score. A one-point increase or decrease in the total score indicated a gain or loss of independence for a specified activity. As there is no pre-specified threshold for the other scales, the threshold was calculated using the Edwards-Nunnaly method, which computes the threshold from the population standard deviation of the total clinical score and the test-retest reliability of the clinical score [47]. The test-retest reliability was estimated by calculating the Cronbach's coefficient alpha of the clinical scales, and was computed using the R package 'psy' [48]. The Edward-Nunnaly thresholds of the different clinical outcomes were submitted to experts (two geriatricians and two nurses) to judge their clinical relevance.

The 'net benefit' (denoted Δ) was defined as the net difference between the number of favourable pairs and the number of unfavourable pairs divided by the total number of pairs [46, 49]. A Δ equal to zero means that intervention does not differ from the control; and Δ ranges from -100%, if all patients in the control group have a score better than all patients in the intervention group, up to 100% in the opposite situation. The Δ was presented with a 95% confidence interval and with the p-value for a test of Δ = 0. This analysis was computed with the R package 'BuyseTest'.

As the distribution of service utilization resulted in many zero scores with a Poisson or negative binomial distribution [50], we used a two-part model, also known as hurdle model. In its first part, the data was considered to be zero versus non-zero, and used a binomial model to

assess the probability of observing a zero value [51]. In the second part, the non-zero observations were modelled with a truncated Poisson or a truncated negative binomial model [51]. As the choice between a zero-augmented Poisson model (ZAP) and a zero-augmented negative binomial model (ZANB) depends on over-dispersion, it was evaluated with a likelihood-ratio test that compared the two models. The hurdle models were computed with the R package 'pscl' [52] and the likelihood-ratio test with the R package 'lmtest' [53].

The exponential coefficients of each outcome were presented for the zero and the count part of each model with their 95% confidence interval (%95 CI) computed by bootstrapping using the package 'Boot' in R software [54, 55]. These coefficients were adjusted using the services evaluated in the six months before inclusion. The exponential coefficient of the zero part can be interpreted as an odds ratio (OR). The exponential coefficient of the count part can be interpreted as a relative risk (RR).

## Results

### Description of the three disability profiles in the intervention group

The scale cut-off showed that 93.99% of the individuals in the 'IADL and initial cognitive impairments (IADL (cogn.))' disability profile had significant limitations in performing IADL (see Table 1). Half the people in this profile also had slight cognitive impairments (i.e. with a CPS score higher than one); and 19.59% had depressive symptoms.

In the 'functional and cognitive impairments (func., cogn.)' disability profile, up to 99.29% presented significant limitations on the IADL, and up to 80.83% on the ADL. Additionally, 96.55% of individuals were cognitively impaired. This group therefore consisted of individuals with significant functional and cognitive limitations.

Finally, the profile called 'functional, cognitive and behavioural problems (func., cogn., behav.)' consisted of individuals who combined functional limitations, cognitive impairments and behavioural problems. Functional limitations were slightly less severe than in the group with functional and cognitive impairments, in which 95.53% of study participants were above the IADL cut-off and 67.89% above the ADL cut-off, although all presented with behavioural problems.

Regardless of the patients' disability profiles, a considerable majority of family carers had a significant psychological burden, the more severe the disability, the higher the proportion of family carers who had a significant burden, whatever their living arrangement (i.e. cohabitant or not).

### Which intervention is most suitable for which disability profile? Patients with IADL limitations and initial cognitive impairments

As Tables 2 and 3 show, basic care coordination was associated with consistent results for patients with 'IADL limitations and initial cognitive impairments' for clinical outcomes and the use of high and low-value services. Indeed, the net benefit (Δ) with regard to the depressive score was 18.90%, [95% CI: 9.58; 27.20], thus significantly favouring the intervention group. This means that there were 18.90% more 'favourable' pairs (i.e. DRS for patients in the intervention group that were larger by at least 2 units than DRS for patients in the control group) than 'unfavourable' pairs. Similar results were observed with regard to the Δ for quality of life (15.70%, [95% CI: 2.44; 29.40]). However, the Δ for the burden on cohabitant family carers was -60.00%, [95% CI: -100; -14.20], thus significantly favouring the control group. For this population subgroup, the control group was favoured in all three categories of CM. The Δ for the justified use of personal nursing care was 12.20% [95% CI: 1.36; 21.20], thus significantly

**Table 1. Proportion of patients with significant difficulties on the clinical scale (% above cut-off) by disability profile.**

| | Intervention group | | Control group | |
|---|---|---|---|---|
| | At enrolment | 6 months after enrolment | At enrolment | 6 months after enrolment |
| **Patients with IADL limitations and initial cognitive impairments** | | | | |
| IADL | 93.99 | 75.15 | 77.41 | 74.24 |
| ADL | 0.31 | 7.91 | 3.99 | 13.85 |
| CPS | 10.62 | 12.93 | 7.93 | 11.75 |
| Behav. | 6.84 | 6.95 | 11.92 | 9.83 |
| DRS | 19.59 | 17.57 | 33.16 | 24.84 |
| Burden non-cohab | 55.14 | 48.12 | 29.23 | 40.59 |
| Burden cohab | 59.93 | 58.94 | 48.00 | 39.31 |
| **Patients with functional and cognitive impairments** | | | | |
| IADL | 99.29 | 98.53 | 99.29 | 100 |
| ADL | 80.83 | 79.48 | 90.48 | 92.29 |
| CPS | 96.55 | 90.37 | 95.12 | 92.73 |
| Behav. | 25.00 | 25.81 | 29.05 | 25.43 |
| DRS | 34.64 | 32.80 | 39.17 | 40.07 |
| Burden non-cohab | 58.82 | 60.75 | 64.26 | 60.47 |
| Burden cohab | 67.20 | 65.89 | 60.70 | 65.22 |
| **Patients with functional, cognitive and behavioural problems** | | | | |
| IADL | 95.53 | 95.45 | 100 | 100 |
| ADL | 67.89 | 71.67 | 73.58 | 72.92 |
| CPS | 90.24 | 88.62 | 81.71 | 87.65 |
| Behav. | 100 | 89.50 | 100 | 88.13 |
| DRS | 58.94 | 57.92 | 69.92 | 67.50 |
| Burden non-cohab | 77.08 | 64.52 | 60.42 | 54.84 |
| Burden cohab | 78.52 | 82.35 | 85.33 | 94.96 |

favouring the intervention group. The probability of using the emergency department services did not differ significantly between the intervention and control groups. Nevertheless, the risk of using these service again was significantly lower (2.63 times) among the users of emergency department services in the intervention group than in the control group (RR = 0.38 [95% CI: 0.21; 0.55]). Finally, the probability of seeing GPs out of hours was significantly lower in the intervention group (OR = 0.39, [95% CI: 0.18; 0.79]).

Low-intensity CM was not associated with positive results except for users of emergency department: in the intervention group, their risk of using this service again was 1.75 times lower (RR = 0.57, [95% CI: 0.41; 0.76]). In the intervention group, the risk of visiting a GP out of hours again was 1.96 times lower (RR = 0.51, [95% CI: 0.31; 0.73]).

High-intensity CM was associated with positive results for different various outcomes. The Δ for ADL, depressive and quality-of-life scores significantly favoured the intervention group (ADL: 18.40%, [95% CI: 14.60; 22.00]), (DRS: 32.90%, [95% CI: 29.10; 36.40]), (QL: 22.50%, [95% CI: 18.60; 27.00]). The Δ for the justified use of personal nursing care was -7.00% [95% CI: -10.60; -2.78], thus significantly favouring the control group. The risk of using the emergency department services again was 2.38 times lower in the intervention group than in the control group (RR = 0.42, [95% CI: 0.34; 0.52]). Those in the intervention group had a significantly lower probability of using an out-of-hours GP either once (OR = 0.38, [95% CI: 0.26; 0.54]) or on one or more subsequent occasions (RR = 0.50, [95% CI: 0.37; 0.67]).

**Table 2. Generalized pairwise comparisons: Results by CM category for patients with IADL limitations and initial cognitive impairments.**

| | Pre-specified threshold | Pairwise probability | | Δ [95% CI] (%) | Δ pvalue |
|---|---|---|---|---|---|
| | | CM> Control (%) | CM < Control (%) | | |
| **Basic care coordination »** | | | | | |
| ADL | 1 | 35.52 | 36.46 | -0.94 [-9.41;7.67] | 0.875 |
| IADL | 1 | 35.15 | 46.19 | -11.00 [-21.50;-0.64] | 0.0417*** |
| DRS | 2 | 37.57 | 18.64 | 18.90 [9.58;27.20] | <0.0001*** |
| QL | 5 | 40.77 | 25.06 | 15.70 [2.44;29.40] | 0.0104*** |
| Burden no cohab | 5 | 26.87 | 42.76 | -15.90 [-35.60;2.39] | 0.125 |
| Burden cohab | 5 | 9.09 | 69.09 | -60.00 [-100;-14.20] | <0.0001*** |
| Nursing care | 1 | 30.86 | 18.70 | 12.20 [1.36;21.20] | 0.0104*** |
| **Low-intensity CM** | | | | | |
| ADL | 1 | 33.12 | 34.78 | -1.65 [-8.66;6.94] | 0.7604 |
| IADL | 1 | 41.12 | 41.63 | -0.52 [-7.33;6.58] | 0.9062 |
| DRS | 2 | 28.16 | 28.21 | -0.06 [-6.56;6.71] | 1 |
| QL | 5 | 34.02 | 28.35 | 5.67 [-3.53;12.60] | 0.2604 |
| Burden no cohab | 5 | 26.03 | 38.95 | -12.90 [-24.40;1.24] | 0.0729 |
| Burden cohab | 5 | 22.93 | 43.33 | -20.40 [-38.20;-4.50] | 0.0104*** |
| Nursing care | 1 | 28.57 | 21.51 | 7.06 [1.27;13.30] | 0.0312*** |
| **High-intensity CM** | | | | | |
| ADL | 1 | 43.80 | 25.41 | 18.40 [14.60;22.00] | <0.0001*** |
| IADL | 1 | 33.17 | 47.85 | -14.70 [-18.80;-10.70] | <0.0001*** |
| DRS | 2 | 44.37 | 11.47 | 32.90 [29.10;36.40] | <0.0001*** |
| QL | 5 | 42.12 | 19.58 | 22.50 [18.60;27.00] | <0.0001*** |
| Burden no cohab | 5 | 33.50 | 33.17 | 0.33 [-5.28;4.71] | 0.9167 |
| Burden cohab | 5 | 25.37 | 42.98 | -17.60 [-27.30;-6.96] | <0.0001*** |
| Nursing care | 1 | 21.63 | 28.60 | -6.97 [-10.60;-2.78] | <0.0001*** |

\*\*\* indicates a significant difference between CM and control

## Patients with functional and cognitive impairments

The results in Tables 4 and 5 show that basic care coordination was not enough to support people with significant functional and cognitive limitations. The only significant Δ for the

**Table 3. Results by CM category of hurdle model for patients with IADL limitations and initial cognitive impairments.**

| | Model | OR [95% CI] | RR [95% CI] |
|---|---|---|---|
| **Basic care coordination** | | | |
| Emergency visits | ZAP | 1.18 [0.52;2.21] | 0.38 [0.21;0.55] *** |
| Out-of hours GP visits | ZAP | 0.39 [0.18;0.79] *** | 0.88 [0.61;1.18] |
| **Low-intensity CM** | | | |
| Emergency visits | ZAP | 1.43 [0.91;2.18] | 0.57 [0.41;0.76] *** |
| Out-of hours GP visits | ZAP | 0.87 [0.49;1.62] | 0.51 [0.31;0.73] *** |
| **High-intensity CM** | | | |
| Emergency visits | ZAP | 0.96 [0.71;1.26] | 0.42 [0.34;0.52] *** |
| Out-of hours GP visits | ZAP | 0.38 [0.26;0.54] *** | 0.50 [0.37;0.67] *** |

\*\*\* indicates a significant difference between CM and control

**Table 4. Generalized pairwise comparisons: Results by CM category for patients with functional and cognitive impairments.**

| | Pre-specified threshold | Pairwise probability | | Δ [95% CI] (%) | Δ pvalue |
|---|---|---|---|---|---|
| | | CM> Control (%) | CM < Control (%) | | |
| **Basic care coordination** | | | | | |
| ADL | 1 | 44.99 | 25.77 | 19.20 [5.27;35.80] | 0.0208*** |
| IADL | 1 | 33.66 | 33.27 | 0.39 [-13.70;17.40] | 0.9479 |
| DRS | 2 | 35.43 | 25.54 | 9.89 [-4.17;23.20] | 0.2188 |
| QL | 5 | 36.98 | 28.12 | 8.85 [-23.50;39.90] | 0.6354 |
| Burden no cohab | 6 | 23.00 | 43.00 | -20.00 [-62.90;18.60] | 0.4271 |
| Burden cohab | 6 | 18.90 | 45.75 | -26.80 [-56.50;3.36] | 0.0833 |
| Nursing care | 1 | 6.48 | 23.66 | -17.20 [-28.20;-8.18] | 0.0104*** |
| **Low-intensity CM** | | | | | |
| ADL | 1 | 41.97 | 28.91 | 13.10 [4.40;20.80] | <0.0001*** |
| IADL | 1 | 37.54 | 31.14 | 6.40 [-1.44;13.20] | 0.0833 |
| DRS | 2 | 33.68 | 27.54 | 6.14 [-3.47;13.10] | 0.0938 |
| QL | 5 | 35.92 | 27.04 | 8.88 [-2.12;18.70] | 0.125 |
| Burden no cohab | 6 | 34.44 | 23.66 | 10.80 [-3.19;26.50] | 0.1771 |
| Burden cohab | 6 | 35.46 | 28.14 | 7.33 [-5.54;19.60] | 0.3229 |
| Nursing care | 1 | 5.63 | 23.77 | -18.10 [-23.80;-14.50] | <0.0001*** |
| **High-intensity CM** | | | | | |
| ADL | 1 | 47.95 | 25.89 | 22.10 [14.20;29.60] | <0.0001*** |
| IADL | 1 | 38.23 | 28.21 | 10.00 [2.74;16.60] | <0.0001*** |
| DRS | 2 | 33.84 | 27.18 | 6.67 [0.69;12.20] | 0.0312*** |
| QL | 5 | 32.03 | 32.26 | -0.23 [-11.20;12.70] | 0.9896 |
| Burden no cohab | 6 | 27.03 | 32.28 | -5.25 [-16.20;8.22] | 0.4271 |
| Burden cohab | 6 | 27.88 | 35.63 | -7.75 [-18.40;4.16] | 0.1354 |
| Nursing care | 1 | 3.66 | 25.38 | -21.70 [-26.40;-16.50] | <0.0001*** |

*** indicates a significant difference between CM and control

intervention group was found for ADL scoring (19.20%, [95% CI: 5.27; 35.80]). The Δ for the justified use of personal nursing care was -17.20%, [95% CI: -28.20; -8.18], thus favouring the control group. No significant difference was found between groups for the use of low-value services.

**Table 5. Results by CM category of hurdle model for patients with functional and cognitive impairments.**

| | Model | OR [95% CI] | RR [95% CI] |
|---|---|---|---|
| **Basic care coordination** | | | |
| Emergency visits | ZAP | 0.99 [0.31;2.67] | 1.23 [0.49;2.25] |
| GP out-of-hours visits | ZAP | 1.16 [0.42;4.77] | 0.89 [0.43;1.94] |
| **Low-intensity CM** | | | |
| Emergency visits | ZAP | 1.07 [0.69;1.66] | 0.68 [0.51;0.87] *** |
| GP out-of-hours visits | ZAP | 0.74 [0.44;1.15] | 0.57 [0.38;0.77] *** |
| **High-intensity CM** | | | |
| Emergency visits | ZAP | 0.86 [0.54;1.32] | 0.61 [0.44;0.80] *** |
| GP out-of-hours visits | ZANB | 0.84 [0.11;1.33] | 0.59 [0.24;1.12] |

*** indicates a significant difference between CM and control

Low-intensity CM was associated with an improvement in functional status. The Δ in ADL scores favoured the intervention group (13.10%, [95% CI: 4.40; 20.80]). Although the percentage of favourable pairs was higher for all other clinical outcomes in the intervention group, the Δ was not significant. Furthermore, the probability of using emergency department services did not differ significantly between the two groups, but, among intervention-group users of emergency departments, the risk they would use subsequent time this service was 1.46 times lower (RR = 0.68, [95% CI: 0.51; 0.87]). Among service users, the risk that people in the intervention group would consult GPs out-of-hours subsequent time was significantly lower (RR = 0.57, [95% CI: 0.38; 0.77]). However, results for the justified use of personal nursing care were similar to those for the basic care coordination profile, with Δ significantly favouring the control group (-18.10%, [95% CI: -23.80; -14.50%]).

High-intensity CM was associated with a significant improvement in functional status within the intervention group, the Δ for ADL scores being 22.10%, [95% CI: 14.20; 29.60]; and the Δ for IADL scores being 10.00%, [95% CI: 2.74; 16.60]. And not only did the Δ for depressive status also favour the intervention group (6.67%, [95% CI; 0.69; 12.20]), the risk that emergency-department services would be used subsequent time was 1.64 lower in the intervention group (RR = 0.64, [95% CI: 0.44; 0.80]) than in the control group. Finally, the Δ for the justified use of personal nursing care was -21.70%, [95% CI: -26.40; -16.50], thus favouring the control group.

### Patients with functional, cognitive and behavioural problems

Basic care coordination among patients with functional and cognitive limitations was insufficient for patients who also had behavioural problems (Tables 6 and 7).

On the other hand, low-intensity CM was associated with a positive improvement in the mental health of older people and their family carers (Tables 6 and 7). Indeed, the Δ significantly favoured the intervention group for the following: depressive score (22.20%, [95% CI: 10.10; 37.40]), burden on the cohabitant FC (28.20%, [95% CI: 4.66; 52.40]), and quality of life (51.40%, [95% CI: 23.10; 83.00]). The Δ for the justified use of personal nursing care was 13.90%, [95% CI: 3.51; 25.10], thus significantly favouring the intervention group. Finally, low-value services did not differ significantly between the groups.

For high-intensity CM, a significant Δ was observed only in the depressive score, favouring the intervention group (13.10%, [95% CI: 2.26; 23.40]) (Tables 6 and 7).

## Discussion

### Interpretation of results

Our results suggest that the impact of CM on patient health, carer and services utilization varies according to the long-term care needs and CM category. So, in choosing the simplest intervention among those that are effective, basic care coordination seems sufficient for patients with IADL limitations and initial cognitive impairments. And, Frail older persons with functional and cognitive impairments or those with additional behavioural problems could benefit from low-intensity CM.

In this study, the main association between case management and clinical outcomes was found for depression and quality of life. This is consistent with a recent systematic review highlighting that CM improves older peoples' psychological health and wellbeing, but not their cognitive or functional status [56].

With regard to the literature [57, 58], we also note associations of CM with the identification of unmet needs and the provision of adequate services. For example, CM showed interesting results for people suffering from functional, cognitive and behavioural problems, whose

**Table 6. Generalized pairwise comparisons: Results by CM category for patients with functional, cognitive and behavioural problems.**

| | Pre-specified threshold | Pairwise probability | | Δ [95% CI] (%) | Δ pvalue |
|---|---|---|---|---|---|
| | | CM> Control (%) | CM < Control (%) | | |
| **Basic care coordination** | | | | | |
| ADL | 1 | 34.03 | 38.19 | -4.17 [-33.20;21.00] | 0.8229 |
| IADL | 1 | 39.93 | 32.47 | 7.47 [-24.20;44.10] | 0.6562 |
| DRS | 3 | 31.42 | 33.51 | -2.08 [-30.40;19.50] | 0.8958 |
| QL | 6 | 46.94 | 22.45 | 24.50 [-42.70;80.70] | 0.4167 |
| Burden no cohab | 6 | 44.00 | 44.00 | 0 [-69.50;75.60] | 1 |
| Burden cohab | 7 | 38.84 | 28.10 | 10.70 [-25.80;52.50] | 0.6354 |
| Nursing care | 1 | 23.61 | 19.44 | 4.17 [-17.60;25.60] | 0.7396 |
| **Low-intensity CM** | | | | | |
| ADL | 1 | 34.84 | 42.42 | -7.58 [-26.60;5.26] | 0.375 |
| IADL | 1 | 36.74 | 34.37 | 2.38 [-11.10;15.80] | 0.75 |
| DRS | 3 | 42.13 | 19.96 | 22.20 [10.10;37.40] | <0.0001*** |
| QL | 6 | 63.52 | 12.10 | 51.40 [23.10;83.00] | <0.0001*** |
| Burden no cohab | 6 | 53.06 | 24.49 | 28.60 [-9.79;66.10] | 0.2083 |
| Burden cohab | 7 | 48.82 | 20.64 | 28.20 [4.66;52.40] | 0.0208*** |
| Nursing care | 1 | 27.86 | 13.98 | 13.90 [3.51;25.10] | 0.0312*** |
| **High-intensity CM** | | | | | |
| ADL | 1 | 41.21 | 35.06 | 6.15 [-7.18;18.10] | 0.3958 |
| IADL | 1 | 32.24 | 35.39 | -3.15 [-17.20;10.50] | 0.7083 |
| DRS | 3 | 34.50 | 21.36 | 13.10 [2.26;23.40] | <0.0001*** |
| QL | 6 | 33.33 | 31.22 | 2.11 [-21.60;24.30] | 0.875 |
| Burden no cohab | 6 | 37.50 | 27.78 | 9.72 [-31.70;59.20] | 0.6667 |
| Burden cohab | 7 | 30.67 | 33.29 | -2.61 [-20.10;12.30] | 0.7292 |
| Nursing care | 1 | 23.39 | 20.51 | 2.88 [-10.40;14.70] | 0.6667 |

*** indicates a significant difference between CM and control

tendency to use support services less may be attributable to a number of factors. For example, their search for formal help may be impeded by the patient's impairments, by the burden on the family carer, by the carer's lack of time; or possibly by the patient's or carer's denial of the need for services or their reluctance to accept help [59]. This result highlights the usefulness of CM by identifying the precise care needs at the beginning of the process of care coordination and supporting the patient's care network for highly disabled older people [60].

**Table 7. Results by CM category of hurdle model for patients with functional, cognitive and behavioural problems.**

| | Model | OR (IC95) | RR (IC95) |
|---|---|---|---|
| **Low-intensity CM** | | | |
| Emergency visits | ZAP | 3.72 [0.92;9.84] | 0.93 [0.18;1.61] |
| GP out-of-hours visits | ZAP | 1.16 [0.37;2.77] | 1.26 [0.46;2.08] |
| **High-intensity CM** | | | |
| Emergency visits | ZAP | 0.86 [0.54;1.32] | 0.61 [0.44;0.80] *** |
| GP out-of-hours visits | ZANB | 0.84 [0.11;1.33] | 0.59 [0.24;1.12] |

*** indicates a significant difference between CM and control

In our study, CM was also associated with reductions not only in the risk that users of the emergency-department services would use these services repeatedly but also in the probability of their GPs using out of hours once or more. Although the literature provides limited evidence that CM for frail older people can reduce the probability of using low-value services [61, 62] some authors have noted in various patient populations that the number of subsequent emergency visits was lower after heterogeneous CM [63, 64].

## Evaluation of CM as a complex intervention

Despite our use of validated statistical methodologies, our results are mixed. Similar findings are shown in the literature. Neither have consistent results been found by classical evaluation of case management as a complex intervention [16, 65]. This lack of consistent results may be explained by three possible factors: (1) the use of various models to deliver case management [16], (2) the difficulties in building a good control group[15], and (3) the non-inclusion of contextual characteristics in the evaluation [66].

With regard to the use of the first possible factor–the use of various delivery models–the 50 projects that implemented case management were grouped in a classification that was based on two criteria: the level of feedback provided to the frail older people's GPs, and the intensity of intervention. This classification made it possible to group projects in a way that obtained the high number of recipients per CM category that is essential to computing statistical analysis. Within the CM categories there was nonetheless some heterogeneity, as these projects were not designed to be standardized–after all, they had been developed by healthcare professionals and had been implemented in various contexts. There were differences with regard not only to project size (i.e. the number of professionals financed), but also to the case managers' profiles (i.e. nurses, psychologists or social workers); the average case load per case manager's fulltime equivalent; and the level of interaction with other care providers [7]. Settings also differed with regard to inter-professional and inter-organizational collaboration. In addition, the information about the activities package was collected at project level, as if a similar CM process had been provided to all people. In our study, for example, while the intensity of the intervention was determined by the average monthly number of home visits, the number of visits might have varied between patients within the same project. CM in projects profiled as 'high-intensity CM' did not mean that all patients received CM at a high intensity. This assumption is confirmed by another indicator of CM intensity [28]: the fluctuations in a case manager's case load between projects in the same category of CM. Per case-manager full-time equivalent, the maximum caseload was 123 patients for low-intensity CM and 77 patients for high-intensity CM, with an upper manageable limit fixed at 40 patients [28].

With regard to the second possible factor, i.e. building a good control group, individuals in each control group shared disability profiles similar to those in the intervention group. For organizational reasons, however, the reference channels used for the intervention groups (i.e. home-care organizations, GP, hospital-based social services, community nurses, and nursing-home waiting-lists) were different than those for the control groups (i.e. home-care organizations). Because, in the control group, we assume that the need for assistance of patients receiving services from home-care organizations to have been well assessed and properly considered, this constituted a risk of selection bias [67]. In addition, the sample size of the control group was relatively small due to resource constraints compared to the size of the intervention group. This required a particular weighting of similar individuals in the control group. With regard to factors such as health conditions and co-morbidities, such weighting may reduce the heterogeneity of profiles in the control group.

With regard to the third possible factor–the non-inclusion of contextual characteristics in our evaluation–our results did not consider broader contextual features. However, the effectiveness of an intervention such as CM may be influenced by the social, organizational and political context in which it is implemented [68]. For example, the implementation of CM is favoured by features of the Belgian health system such as the short notice at which a GP can be consulted, and the low cost to patients of basic care at home [69]. On the other hand, certain features may make its implementation difficult. For example, the principle of the patient's freedom of choice makes it difficult to work with fixed collaborators [69]. Most healthcare providers are paid according to a fee-for-services system, thus favouring the amount of care rather than the collaboration between professionals that is necessary to ensuring continuity and comprehensive care in complex situations [69]. As there is no formal training for case managers in Belgium, their function is often confused with that of other types of practitioner, giving them little legitimacy. There are further inadequacies with regard to the culture of evidence-base practice and the data-sharing tools between healthcare providers. Finally, irrespective of the general characteristics of the Belgian health system, local realities differ widely–an especially important factor, as CM may be more effective when it is tailored to local context than when it is fully standardized [70].

## New paradigm of evaluation

Due to the mixed results of this evaluation and the complexity of the system in which CM may be implemented, we feel it relevant to open a debate on a new evaluative paradigm [71]. This paper has evaluated CM in a standardized form–a black box that is shared by the projects belonging to the same CM category. A systems approach would suggest that we should shift our vision of evaluation and consider CM not as complex interventions but more as events–in this case a set of interconnected actions–that are introduced into a complex system [72]. As these events have the potential to modify the system dynamic, [72] the functions and process of change could be standardized in all projects, while the forms of the interventions might be adapted to different contexts [18].

The evaluation of events intended to change complex systems starts by studying and understanding the situation in terms of a system [72]. This requires a shift from a linear causal logic to new ways of thinking that permit non-linearity, unintended effects, interactions between micro and macro levels, and a dynamic vision of the evolution of individual needs, intervention and context [18, 72].

The main difficulty with such a new paradigm is that the results it obtains remain far removed from political expectations. And indeed, the aim of this new paradigm is no longer to identify a standardized intervention with clear components that can be scaled up everywhere, but to allow a continual process of learning and experimenting with the different stakeholders to create the conditions to bring about change [73].

## Conclusion

To fulfil the policymakers' request that we evaluate the effectiveness of CM, we used an evaluative methodology appropriate to complex interventions. We propose a robust methodology that allows the following: the standardization of the interventions into clear-cut categories, each defined by explicit components; the stratification of the study population in sub-groups, each with similar assistance needs; the building of a comparison group; and the evaluation of the effectiveness of CM for a wide variety of outcomes.

Due to the methodological limitations inherent to applying conventional statistics to the evaluation of bottom-up and complex interventions, the results of this approach will not

provide policymakers with a simple, univocal message. But the methodological limitations in question open the debate on the need to evolve towards other evaluative designs, and suggest that an intervention should be considered as events introduced into a complex system. This approach will be tested in Belgium for the evaluation of integrated care projects.

## Supporting information

**S1 Table. Description of the number of patients by CMI category and by disability profile, and description of the number of projects, by CMI category.**
(DOCX)

**S2 Table. Evaluation of balance of the BelRAI covariables between study arms, by disability profile, for the total sample of "Protocol 3."**
(DOCX)

## Acknowledgments

We are grateful to the Protocol 3 project workers for their participation in the scientific evaluation; to the National Institute of Health and Disability Insurance for their support and interest in finding alternatives to institutionalization; to David Alexander for his precious proofreading of English; and to the Protocol 3 scientific team members, who comprise Anja Declercq, Johanna De Almeida Mello, Sibyl Anthierens, Maja Lopez-Hartmann and Roy Remmen for their advice and their participation in this evaluation.

## Author Contributions

**Conceptualization:** Anne-Sophie Lambert, Sophie Cès, Thérèse Van Durme, Jean Macq.

**Formal analysis:** Anne-Sophie Lambert, Catherine Legrand.

**Methodology:** Anne-Sophie Lambert, Catherine Legrand, Jean Macq.

**Writing – original draft:** Anne-Sophie Lambert, Catherine Legrand, Sophie Cès, Thérèse Van Durme, Jean Macq.

**Writing – review & editing:** Anne-Sophie Lambert, Catherine Legrand, Sophie Cès, Thérèse Van Durme, Jean Macq.

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
