## [Decision Letter · Decision Letter 0]

2 Aug 2019

PONE-D-19-19169

Evaluating case management as a complex intervention: lessons for the future

PLOS ONE

Dear Mrs Lambert,

Thank you for submitting your manuscript to PLOS ONE. After careful consideration, we feel that it has merit but does not fully meet PLOS ONE’s publication criteria as it currently stands. Therefore, we invite you to submit a revised version of the manuscript that addresses the points raised during the review process.

We would appreciate receiving your revised manuscript by Sep 16 2019 11:59PM. To enhance the reproducibility of your results, we recommend that if applicable you deposit your laboratory protocols in protocols.io, where a protocol can be assigned its own identifier (DOI) such that it can be cited independently in the future. For instructions see: http://journals.plos.org/plosone/s/submission-guidelines#loc-laboratory-protocols

We look forward to receiving your revised manuscript.

Kind regards,

Lars-Peter Kamolz, M.D., Ph.D., M.Sc.

Academic Editor

PLOS ONE

Journal Requirements:

Reviewers' comments:

Reviewer's Responses to Questions

**Comments to the Author**

1. Is the manuscript technically sound, and do the data support the conclusions?

Reviewer #1: Partly

Reviewer #2: Yes

2. Has the statistical analysis been performed appropriately and rigorously? 

Reviewer #1: I Don't Know

Reviewer #2: Yes

3. Have the authors made all data underlying the findings in their manuscript fully available?

Reviewer #1: No

Reviewer #2: Yes

4. Is the manuscript presented in an intelligible fashion and written in standard English?

Reviewer #1: No

Reviewer #2: Yes

5. Review Comments to the Author

Reviewer #1: Dear authors, this manuscript tries to highlight a study but it was difficult reading it.

For example in line 69 the authors mentioned 50 Projects, but it was not explained which 50 Projects. Then in line 129 six case studies were mentioned. I lost the overview. How many patients were enrolled?

All in all I have to conclude that the manuscript is not worth for publication.

Reviewer #2: This is a well conducted mixed methods study on a very innovative style and a comprehensive approach to an increasingly relevant problem of frail people. The methodology is described well and in detail.

6. PLOS authors have the option to publish the peer review history of their article (what does this mean?). If published, this will include your full peer review and any attached files.

Reviewer #1: No

Reviewer #2: Yes: Aneesh Basheer

---

## [Author Response · Author response to Decision Letter 0]

11 Sep 2019

Response to reviewers

Dear reviewers and editor, 

Thanks a lot for the smooth process of reviewing our article. Hereafter, you will find response to the two reviewers’ comments. 

We remain, of course, ready to clarify further our work, would they be any other queries.

Warm regards, 

Anne-Sophie Lambert

Reviewer #1: Dear authors, this manuscript tries to highlight a study but it was difficult reading it.

For example in line 69 the authors mentioned 50 Projects, but it was not explained which 50 Projects. Then in line 129 six case studies were mentioned. I lost the overview. How many patients were enrolled?

All in all I have to conclude that the manuscript is not worth for publication.

I would like first thank you for your pertinent comment. I totally agree with your comment. I include different elements in the paper to clarify the lines 69 and 129. 

First in the introduction, I justified the number of projects (line 63 and 69). 75 projects were included in the Protocol 3 programme. Among them, 50 projects implemented Case Management. The others delivered psychological support, occupational therapy or night care. This paper focuses on the 50 projects that implemented Case Management. 

Second in the methodology, I added precisions and references who justified the choice to analyze only six projects in-depth (line 113 - 125). The classification of CM projects started by the building of a normative grid: a list which captures what the stakeholders of the projects believed to be the key components of the projects [1]. To reach this objective, we used multiple embedded case study. This qualitative approach focuses on a small number of cases to highlight a causal relation assumed to be present in a large number of cases [2]. The cases are selected to extract as much diversity as possible regarding the projects‘ characteristics [2, 3]. Thus, we used six in-depth cases studies, selected among the 50 projects that implemented CM. These six projects were selected on the basis of (1) their diversity (i.e. the profile of the case manager (i.e. nurse, occupational therapist, psychologist or social worker)); (2) their geographical location (urban or rural); and (3) their collaboration with an existing organization that, among other criteria, coordinated care. 

Third, in the first version of the manuscript, the number of patients enrolled was included in the S1 Appendix. However, I agree with you, it is necessary to include this information in the manuscript. So, I added the number of projects and the number of patients included in each CM category in the manuscript (lines 143, 145, 147). The total number of beneficiaries and the total number of controls were corrected (line 170). And the reference to S1 Appendix was added when I presented the number of beneficiaries per CM category (line 202, 203, 204). 

Reviewer #2: This is a well conducted mixed methods study on a very innovative style and a comprehensive approach to an increasingly relevant problem of frail people. The methodology is described well and in detail.

 I would like thank you a lot for your positive comment.

---

## [Decision Letter · Decision Letter 1]

10 Oct 2019

Evaluating case management as a complex intervention: lessons for the future

PONE-D-19-19169R1

Dear Dr. Lambert,

We are pleased to inform you that your manuscript has been judged scientifically suitable for publication and will be formally accepted for publication once it complies with all outstanding technical requirements.

With kind regards,

Lars-Peter Kamolz, M.D., Ph.D., M.Sc.

Academic Editor

PLOS ONE

Additional Editor Comments (optional):

Reviewers' comments:

Reviewer's Responses to Questions

**Comments to the Author**

1. If the authors have adequately addressed your comments raised in a previous round of review and you feel that this manuscript is now acceptable for publication, you may indicate that here to bypass the “Comments to the Author” section, enter your conflict of interest statement in the “Confidential to Editor” section, and submit your "Accept" recommendation.

Reviewer #1: All comments have been addressed

Reviewer #2: All comments have been addressed

2. Is the manuscript technically sound, and do the data support the conclusions?

Reviewer #1: Yes

Reviewer #2: Yes

3. Has the statistical analysis been performed appropriately and rigorously? 

Reviewer #1: I Don't Know

Reviewer #2: Yes

4. Have the authors made all data underlying the findings in their manuscript fully available?

Reviewer #1: Yes

Reviewer #2: Yes

5. Is the manuscript presented in an intelligible fashion and written in standard English?

Reviewer #1: (No Response)

Reviewer #2: Yes

6. Review Comments to the Author

Reviewer #1: Dear authors, thank you for adressing all issues in the revised manuscript. I do not have any further comments.

Reviewer #2: Dear authors

You have touched upon a very complicated intervention and tried to evaluate it in the conventional method. That partly explains the complexity of the results and why it may seem very confusing. However, as you rightly mention, currently policy makers are not aware of other approaches to evaluate complex interventions. To that end your paper is very important in not ony conveying the results of the evaluation but also sending a message that newer approaches to evaluation are needed to address such interventions.

7. PLOS authors have the option to publish the peer review history of their article (what does this mean?). If published, this will include your full peer review and any attached files.

Reviewer #1: No

Reviewer #2: Yes: Aneesh Basheer

---

## [Editor Report · Acceptance letter]

21 Oct 2019

PONE-D-19-19169R1 

Evaluating case management as a complex intervention: lessons for the future 

Dear Dr. Lambert:

I am pleased to inform you that your manuscript has been deemed suitable for publication in PLOS ONE. Congratulations! Your manuscript is now with our production department. 

With kind regards,

on behalf of

Dr. Lars-Peter Kamolz 

Academic Editor

PLOS ONE